# Physical Exercise to Improve Functional Capacity: Randomized Clinical Trial in Bariatric Surgery Population

**DOI:** 10.3390/jcm11154621

**Published:** 2022-08-08

**Authors:** María José Aguilar-Cordero, Raquel Rodríguez-Blanque, Cristina Levet Hernández, Javiera Inzunza-Noack, Juan Carlos Sánchez-García, Jessica Noack-Segovia

**Affiliations:** 1Andalusian Plan for Research, Development and Innovation, CTS 367, 18014 Granada, Spain; 2Department of Nursing, Faculty of Health Sciences, University of Granada, 18016 Granada, Spain; 3Andalusian Plan for Research, Development and Innovation, CTS 1068, 18014 Granada, Spain; 4Department of Nursing, University Santo Tomás, Talca 3460000, Chile; 5General University Hospital of Alicante Doctor Balmis, 03010 Alicante, Spain

**Keywords:** obesity, six-minute walking test, bariatric surgery, functional capacity, weight loss

## Abstract

Background: Bariatric surgery is a safe and effective method to lose weight over time. However, some patients fail to achieve healthy weight losses. We aimed to determine if a moderate-intensity physical exercise intervention in patients who underwent bariatric surgery increases their functional capacity thus improving bariatric surgery results. Methods: We conducted a parallel-group non-blinded randomized controlled trial at a surgery clinic in Talca, Chile. A total of 43 participants with obesity and scheduled bariatric surgery completed the six months follow-up. A physical exercise program was conducted in exercise group participants one month after bariatric surgery. Walked distance in the six-minute walk test, BMI, Borg scale of perceptive exertion results and cardiovascular variables were evaluated. Results: Patients’ weight significantly decreased after bariatric surgery but there was no difference between the groups of study. The exercise group progressed from a base value of 550 ± 75 m walked in the six-minute walk test to a sixth-month value of 649.6 ± 68.5 m (*p* < 0.05), whilst the control group yielded base values of 554.4 ± 35.1 and a sixth-month walked distance of 591.1 ± 75.34 (*p* > 0.05). Conclusions: Physical exercise in obese patients undergoing bariatric surgery increased functional capacity independently of weight losses resulting from bariatric surgery.

## 1. Introduction

According to the WHO, being overweight or obese is defined as an abnormal or excessive fat accumulation that may impair health [1]. Body mass index (BMI) is a measure that enables the screening and diagnosis of obesity. It is calculated by dividing a person’s weight in kilograms by the square of the height in m (kg/m^2^). A person with ≥30 BMI is considered obese whilst BMI ≥ 25 and <30 is considered overweight. In 2016, almost 40% of the total population in Chile were overweight and another 34.4% were obese. Obesity and overweight related unhealthy behaviours are top causes of chronic diseases in Chile [2]. According to data from the Global Health Observatory, medium BMI amongst adults in Chile in 2016 was 28 kg/m^2^, being the highest amongst Latin American countries.

Medical treatments aiming to improve these pathologies and their comorbidities have not yielded adequate results in terms of weight loss [3]. However, bariatric surgery (BS) has proven to be an effective method to lose weight over time, being widely accepted by health professionals [4,5,6,7,8]. Moreover, it is regarded as a safe, effective and replicable technique with few side effects which might decrease life quality of patients [9,10]. Among patients undergoing BS there are cases who have failed to achieve healthy weigh losses (≥50% loss of excessive weight) [11]. This fact has predisposed researchers to study these individual cases considering some covariates such as previous BMI, lifestyle, comorbidities and functional capacity [12,13].

Physical functional capacity is defined as the ability of a person to conduct activities on a daily basis [14]. The six-minute walk test (6MWT) constitutes an easy and economic tool for evaluating functional capacity [15,16]. This field test is an excellent mortality/morbidity indicator commonly used to evaluate functional status and response to treatments in patients with cardiac or respiratory disorders [17]. 6MWT is often used to evaluate populations with advanced age and limited mobility [18,19,20] as well as people with severe impairments [21]. Studies conducted to date have shown an association between higher 6MWT walked distance during the preoperative stage and early onset weight loss after surgery [22,23]. In morbidly obese patients, 6MWT distance results tend to decrease and dyspnoea and musculoskeletal pain are more frequent than in non-obese patients, with a negative association between higher BMI and 6MWT results [21]. Bariatric surgery increases 6MWT walked distance [24,25,26,27], and physical activity in conjunction with BS has been suggested to improve life quality and lower mortality rates [28].

The objective of the present study is to determine if a moderate physical exercise intervention in patients who underwent bariatric surgery further improves functional capacity thus enabling patients to achieve healthy weight losses.

## 2. Materials and Methods

A non-blinded randomized clinical trial was conducted in patients who underwent sleeve gastrectomy from 2015 to 2019 in the surgery consultations of the Lircay clinic in Talca, Chile. Patients were randomly allocated to two groups: Experimental group (EG) and control group (CG) (1:1). Randomization was performed using a random permuted block with random block size to avoid predictability due to the non-blinded allocation of participants. Inclusion criteria consisted of obese patients aged between 18–60 years old with no absolute contraindications to practice physical exercise [23], awaiting bariatric surgery, with weight <180 kg and residency in Talca city. Exclusion criteria consisted of pregnancy, severe pathologies that might compromise trial participation, and lack of adherence to the study. Included participants signed the informed consent and the study protocol was applied afterwards. Anamnesis and anthropometric variables were collected after enrolment. Participants were offered a 24 weeks moderate physical exercise program starting the first month after the BS operation. The intervention consisted of performing aerobic exercises along with muscular strengthening exercises three times a week, with one and a half hours for each session. EG patients’ variables were monthly assessed up to the sixth month after bariatric surgery whilst CG patients were evaluated the first, and the sixth month after BS operation according to clinical practice. 6MWT was performed before, one month, and six months after BS in both groups. This study was conducted in accordance with the Declaration of Helsinki and was approved by Ethics Committee of the University Santo Tomás of Santiago de Chile, Chile. This research was retrospectively registered in ClinicalTrials.gov in 2017 with the number “NCT03159312”. Results obtained in this research are reported following the CONSORT guidelines for clinical trials.

### 2.1. Sample Size Calculation

Sample size was calculated from 6MWT distance results obtained in another study performed in a BS population [26]. Therefore, values of 50 m with 45 m of standard variation were proposed as the minimum difference required to obtained statistically significant values before and after surgery. Calculations yielded a minimum of 17 participants for both groups considering a significance level of 0.05 with 90% statistic power. The chosen sample size was increased by 20% in order to allow for potential follow-up losses. This resulted in a required total of 41 participants. Despite sample size calculations, we hypothesized that the physical training protocol would increase 6MWT differences after BS.

### 2.2. Characteristics of Participants

Assessed anthropometric variables were weight (kg) and height (m). Assessments were performed between 9–11 a.m. wearing light clothing and no footwear. BMI was calculated for each participant. Collected sociodemographic data consisted of sex, age, education level and marriage status.

### 2.3. Exercise Protocol

Preparatory series of exercises were performed one week before the physical exercise program was conducted. These series consisted of repetitions of exercises performed at an increasing pace until reaching the aimed intensity. A physiotherapist monitored the entire process.

Patients walked at 54% of their physical capacity and frequency resistance in the walking treadmill (HP Cosmos^®^, Nussdorf-Traunstein, Germany). Training intensity was increased to 59% until finishing the aerobic training season during 30 min in the treadmill. Biceps, triceps, deltoids and pectoral muscle strengthening exercises were performed progressively in terms of intensity and number of repetitions following one-repetition maximum text (1RM) [29]. Training in cycle ergometer (Monark^®^ model 894e, Vansbro, Swede) was carried out without additional weight during 15 min. If the patient was not able to perform at least 10 min of continuous exercise, duration was divided into two/three different periods along with strengthening or stretching exercises. In order to finish the series, upper and lower body stretching exercises as well as respiratory exercises were performed (inspiration, deep expiration, and diaphragmatic respiration).

Functional capacity was assessed using 6MWT according to the American Thoracic Society (ATS) [23]. Participants were gathered in the morning equipped with sport clothing. Hemodynamic parameters were measured, namely blood pressure (B/P) and heart rate (H/R) (after previous ten-min repose). The Borg scale of perceptive exertion [30] was applied both at the beginning and at the end of the test. Participants were told to walk as fast as possible through a 30 m long hall marked each 5 m. Participants were also suggested to stop the test if they felt indisposed.

Measurements were taken by trained staff. Patients’ weight was assessed 72 h after the last training session in the same order and by the same trained staff to avoid miscalculations due to water losses. Any other kind of physical activity programme was not allowed throughout the study.

### 2.4. Statistical Analysis

Base-value measures in both groups were analysed using Student *t* test. *p*-values < 0.05 were considered statistically significant and homogeneity of sample base values was analysed. Variables were statistically described as follows: Continuous and categorial variables frequencies were reported as mean ± standard deviation and median. The Shapiro–Wilk test was used to evaluate normality of variables. Variance and sphericity were evaluated using Levene and Mauchly tests respectively. We used two way ANOVA with repetitive measurements to evaluate time-group interactions. When the F value was found significant, we applied post hoc Bonferronic test to measure differences between mean values. *p* < 0.05 was considered statistically significant. We conducted data analysis using SPSS v23.0 (IBM Corp., New York, NY, USA).

## 3. Results

Two patients left the study being follow-up losses (4.4%). 43 subjects completed the study, 21 were allocated to EG group (6 men and 15 women) and 22 were allocated to CG (5 men and 17 women). EG group attendance at training sessions was 97%. All participants in both groups attended 100% of the evaluations. The 43 participants were included in all analyses. A flow diagram of participants is presented in Figure 1.

Sociodemographic characteristics of participants are presented in Table 1. Overall, marriage status and education level were significantly different across groups. Sex distribution and mean age were similar in both groups. However, two thirds of the participants were women.

Anthropometric characteristics of participants are presented in Table 2. The preoperatory weight of patients was 95.66 ± 13 in the EG group and 103.44 ± 14.4 in the CG group. At six months, it was 69.85 ± 9.16 kg and 68.409 ± 11.31 kg in EG and CG, respectively. Base BMI was 35.5 ± 3.3 and 36.7 ± 3.3 in EG and CG, respectively. At six months, it was 26.05 ± 2.95 and 24.32 ± 3.16 in EG and CG, respectively.

6MWT was performed by every participant normally. None of the participants showed any health issues or pathological symptoms throughout the test. When comparing 6MWT results in EG and CG groups, walked base values were lower in EG with a mean difference of 4.46 m. Sixth month evaluation showed an increase in the walked distance from preoperatory stage of 99.76 and 36.59 m for EG and CG, respectively (Figure 2).

ANOVA of repeated measures revealed significant time-group interactions for BMI (F = 14.60; *p* < 0.0001) and 6MWT (F = 20.44; *p* < 0.0001). Multiple comparisons results are presented in Table 3.

### 3.1. Intragroup Multiple Comparisons

Experimental group multiple comparisons showed significant differences for BMI and 6MWT distances between base values and one-month values (*p* < 0.0001), base values and six months values (*p* < 0.0001) and one-month values and six months values (*p* < 0.0001). BMI control group comparisons also showed significant differences between base values versus one month (*p* < 0.0001), base values versus six months (*p* < 0.0001) and one month versus six months (*p* < 0.0001). When analysing CG 6MWT distance results, significant differences were evident in this group for base values versus one-month values (*p* = 0.0098) and base values versus six months (*p* < 0.0001). However, no significant differences were observed between one month and six months values in 6MWT results (Table 3).

### 3.2. Intergroup Multiple Comparisons

Comparisons between groups (experimental versus control) showed that only six min-walk performance at six months was significantly different between groups (*p* = 0.0003). Following this trend, base, one month and sixth month BMI values were not significantly different between EG and CG (Table 3).

Regarding cardiovascular parameters, heart rate (HR) in EG after 6MWT varied from 141.75 beats per minute (BPM) to 138.92 BPM from one month to the sixth month. Nonetheless, this difference was not statistically significant (*p* = 0.09). Systolic and diastolic pressure values assessed after 6MWT performance were significantly lower six months after operation (*p* <0.001). CG ANOVA showed significant values for all variables with the exception of systolic blood pressure after performing the test (*p* = 0.06) (Table 4). Lastly, reported values of perceived exhaustion after 6MWT were significantly lower following BS, yet no statistically significant difference was observed between both groups. 

## 4. Discussion

The present open-label randomized controlled trial was designed to analyse the influence of a mild physical exercise program on the functional capacity of participants who underwent bariatric surgery assessed using the 6MWT. Overall, 6MWT walked distance values were approximately 20% lower than those reported in the literature about non-morbid obese patients with similar age [31,32] as was expected given the patients’ condition. BMI differences were not significant between the groups of study. Nonetheless, the walked distance in the 6MWT improved significantly among EG participants compared to CG participants at one month and six months after BS which implies an improvement in functional capacity. 

Bariatric surgery has proven to be an excellent treatment for obesity and related pathologies by means of reducing patients’ weight, allowing them to introduce physical exercise into their routine and improving functional capacity [33,34]. Nonetheless, BS patients tend to regain weight after the procedure over the years [33,34,35], meaning that it is interesting to apply other methods to enhance results associated to BS and break the vicious circle. A meta-analysis on this subject suggested that cardiorespiratory fitness is a more powerful predictor of cardiovascular disease than BMI [36]. Other authors coincide in that cardiorespiratory fitness is a crucial confounder when assessing health benefits regarding interventions designed to prevent obesity [37]. Hence, interventions designed to improve physical capacity could be beneficial to reinforce BS results in the long term. However, there is a strong necessity of endorsing physical activity and exercise throughout the healthcare system [38]. 

Observed improvements in 6MWT results obtained by control group participants are in line with those obtained by other authors [24,25,26,27]. Bariatric surgery has proven to be an effective method to enable patients to perform physical activities. Nonetheless, base 6MWT walked distances in the present study were significantly higher in comparison to other studies [24,25,26,27]. Higher base performance in our study could be attributed to encouragement from the research team to participants as well as lower base BMI. Body weight and thigh diameter have important implications in 6MWT results [39,40].

Functional capacity in terms of walked distance in the 6MWT improved significantly in the experimental group in comparison to the control group six months after BS. Patients who kept practicing mild exercises showed a significant increase of 99.76 m in the distance walked. On the other hand, CG patients showed an increase of only 36.59 m from base to the sixth month after surgery values. Patients were not allowed to practice any other physical training activity thorough the study. This difference between both groups can be attributed to quadriceps muscular strengthening in EG participants by means of practicing resistance exercises. 

In line with our results, another author evaluated the influence of physical exercise in BS results yielding significant improvements in health outcomes [34]. Stegen et al. conducted a study in BS patients who took part in a physical exercise program four months after the operation. This resulted in an increase in the distance walked by both intervention and control groups with a higher increase in the distance walked by the exercise group which was attributed to physical status improvement, and higher aerobic capacity with lower HR and diastolic pressure due to cardiac modulation [41]. Contrasting with these results, we did not find significant differences between EG and CG in any of the cardiovascular parameters evaluated. However, bariatric surgery significantly improved these results in both groups. An exception was constituted by systolic blood pressure in the CG after 6MWT performance which was not statistically significant. These results can be attributed to weight losses. Heart rate decrease in the exercise group can be attributed to autonomous regulation which has important implications since it is considered as a mortality predictor [42]. A meta-analysis on physical training interventions to improve the outcomes of BS concluded that exercise training programs performance after BS are effective to enhance physical fitness, yet no effect on blood pressure was concluded [12]. Only one reviewed paper specifically evaluated sleeve gastrectomy patients taking part in a balanced training protocol [12]. Thus, we consider that our work confirms the benefits of physical exercise practice in these patients and further contributes to the body of literature. Finally, values obtained in HR were similar to those obtained by other authors [32,41].

### Limitations of the Study

Most of our patients were women. This difference could imply bias due to the characteristics of the female sex with lower ability to perform physical exercises and, therefore, potentially decreased results in 6MWT. 

Our findings might not be extrapolated to patients with higher ranges of BMI. Higher BMI values imply greater challenges when participating in mild physical exercise programs. 

The mild physical activity program was based on exercises performed in a population with physical restrictions as well as patients with arthritis and limited air flow. Due to heterogeneity, our results could not be directly compared with other studies.

## 5. Conclusions

We conclude in the present research that bariatric surgery, in addition to a mild physical exercise program, improves health condition in patients by means of normalising health parameters as well as increasing functional capacity of trained patients. Functional capacity was measured via 6MWT which has proven to be an assessment tool with low cost and easy applicability as well as a predictor of health status in patients undergoing bariatric surgery. 

## Figures and Tables

**Figure 1 jcm-11-04621-f001:**
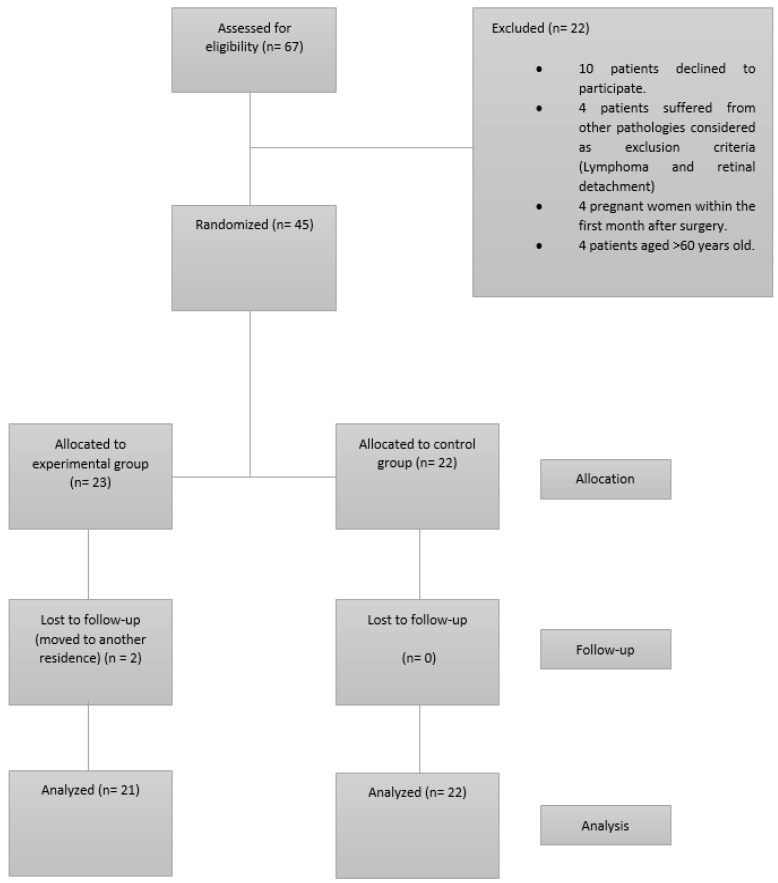
Flow diagram.

**Figure 2 jcm-11-04621-f002:**
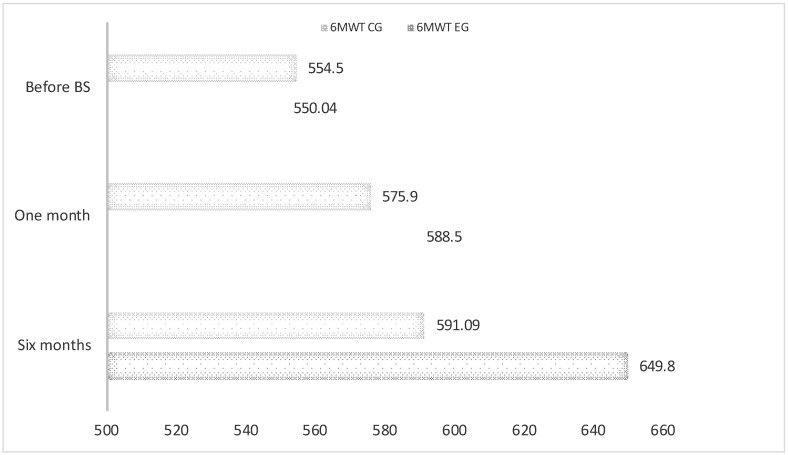
6MWT mean walked distances. Walked distances showed in m.

**Table 1 jcm-11-04621-t001:** Sociodemographic characteristics of participants.

Variable	Experimental Group (EG)	Control Group (CG)
Sex		
Male/Female	6/15	5/17
Age	37.83 ± 7.43	35.09 ± 4.84
Education level		
Primary or secondary education	0	4 (18.2%)
Certificate of higher education	3 (25%)	4 (18.2%)
Bachelor degree or higher education	18 (75%)	14 (63.6%)
Marriage Status		
Married	14 (58.3%)	6 (27.3%)
Divorced	3 (25%)	0
Single	4 (16.7%)	16 (72.7%)

Sociodemographic characteristics of participants. Age values are presented as mean ± standard deviation and other values are presented as absolute or relative frequency.

**Table 2 jcm-11-04621-t002:** Weight and BMI of participants.

Variable	Group	Preoperatory Stage	One Month	Six Months
Weight	EG group	95.66 kg (±13.07)	83.143 kg (±11.35)	69.85 kg (±9.16)
	CG group	103.04 kg (±14.40)	88.273 kg (±14.63)	68.409 kg (±11.31)
BMI	EG group	35.52 (±3.34)	30.95 (±3.26)	26.05 (±2.95)
	CG group	36.73 (±3.31)	31.41 (±3.75)	24.32 (±3.16)

Weight and BMI are provided as mean ± standard deviation.

**Table 3 jcm-11-04621-t003:** Intragroup and Intergroup Multiple Comparisons.

	Comparison	Mean Differences	95% IC	*p* Value
Intragroup BMI	Base CG v/s CG one month	−5.193	−6.138 to −4.248	<0.0001
	Base CG v/s CG six months	−10.10	−11.04 to −9.153	<0.0001
	CG one-month v/s CG six months	−4.905	−5.850 to −3.960	<0.001
	Base EG v/s EG one month	−5.867	−6.790 to −4.944	<0.001
	Base EG v/s EG six months	−12.96	−13.98 to −12.04	<0.001
	EG one-month v/s EG six months.	−7.091	−8.014 to −6.168	<0.001
Intergroup BMI	Base CG v/s Base EG	1.31	−1.323 to 3.584	0.79
	CG one-month v/s EG one month	0.458	−1.997 to 2.910	0.999
	CG six months v/s EG six months.	−1.729	−4.183 to 0.7243	0.269
Intragroup 6MWT walked distances	Base CG v/s CG one-month.	−38.07	−55.44 to −20.69	<0.001
	Base CG v/s CG 6 months	−99.35	−116.7 to −81.98	<0.001
	CG one-month v/s CG six months.	−61.29	−78.66 to 43.91	<0.001
	Base EG v/s EG one-month	−21.36	−38.34 to −43.91	0.0098
	Base EG v/s EG six months	−36.55	−53.52 to −19.57	<0.001
	EG one-month v/s EG six months.	−15.18	−32.15 to 1.791	0.0890
Intergroup 6MWT walked distances	Base CG v/s Base EG	−4.136	−39.55 to 31.38	0.9890
	CG one-month v/s EG one month	12.57	−22.85 to 47.98	0.7753
	CG six months v/s EG six months.	58.67	23.26 to 94.09	0.0003

**Table 4 jcm-11-04621-t004:** 6MWT cardiovascular parameter results.

Variable	EG Preoperatory	EG One-Month	EG Six-Months	*p* Value	CG Preoperatory	CG One-Month	CG Six-Months	*p* Value
Heart rate (Beats/min)								
Base	84.00 ± 14.5	88.17 ± 11.2	73.75 ± 12.5	<0.001	82.09 ± 11.9	80.5 ± 14.3	75.0 ± 15.3	<0.001
After 6MWT	141.75 ± 14.7	145.92 ± 21.9	138.92 ± 19.9	0.09	141.55 ± 18.6	145.82 ± 19.8	133.55 ± 28.0	<0.001
Base blood pressure								
Systolic	120.58 ± 6.2	109.08 ± 5.2	109 ± 8.4	<0.001	122.18 ± 16.9	117.18 ± 14.1	109.73 ± 14.1	<0.001
Diastolic	82.33 ± 12.4	74.00 ± 5.3	70.92 ± 8.1	<0.001	84.73 ± 15.45	74.73 ± 9.10	76.55 ± 8.7	<0.001
Post-6MWT blood pressure								
Systolic	132.83 ± 13.2	122.75 ± 11.1	115.08 ± 11.7	<0.001	136.45 ± 20.7	126.64 ± 15.2	114.45 ± 11.9	0.06
Diastolic	88.17 ± 6.6	80.25 ± 2.3	72.08 ± 7.4	<0.001	88.64 ± 15.8	79.64 ± 9.5	77.73 ± 10	<0.001

Data presented as mean ± standard deviation. Significance level *p* < 0.05.

## Data Availability

Data regarding the study are available upon request to the corresponding author.

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
