# Peer review of "Physical Exercise to Improve Functional Capacity: Randomized Clinical Trial in Bariatric Surgery Population"

_jcm, 2022, doi:10.3390/jcm11154621_

Round 1

Reviewer 1 Report

Aguilar-Cordero et al investigated the effect of physical exercise on the functional capacity of patients who underwent bariatric surgery (BS). The main finding of the study is that exercise improves functional capacity of BS patients, assessed by six-minute walk test. The methodology is sound and the manuscript has a clear message that is in line (in general) with the findings of previous similar studies. However, a few issues should be addressed before publication.

Major points:

- The authors mention in the abstract “Borg scale of perceptive exertion results and cardiovascular variables were evaluated” but the Borg scale results are not included in the manuscript.

-The discussion of the results should be more focused and directed towards the comparison of the current findings with previous studies.

-  What type of BS was performed? Are there differences between the two groups in terms of BS type?

- Some parts of the text should be rewritten for clarity (for example lines 92-94 and lines 261-263). Moreover, the conclusion section should be focused on the main finding of the study (increased functional capacity in the training group) and do not extrapolate into other health parameters or morbidity risk.

Minor point:

- BS has been shown to improve metabolic parameters and is considered as a therapeutic tool for diabetes remission. Have you studied the metabolic parameters of the patients included in your cohorts to find out if exercise had a positive influence?

Author Response

  • Aguilar-Cordero et al investigated the effect of physical exercise on the functional capacity of patients who underwent bariatric surgery (BS). The main finding of the study is that exercise improves functional capacity of BS patients, assessed by six-minute walk test. The methodology is sound and the manuscript has a clear message that is in line (in general) with the findings of previous similar studies. However, a few issues should be addressed before publication.

We thank the reviewer for his/her input on our manuscript aimed to improve its clarity.

Major points:

  • - The authors mention in the abstract “Borg scale of perceptive exertion results and cardiovascular variables were evaluated” but the Borg scale results are not included in the manuscript.

As pointed by the reviewer, no mention to the Borg scale results was made throughout the manuscript. This is a significant omission. Although values of perceived exertion were lower after BS, we found no difference between groups of study. We have pointed this aspect in section 3.2. We consider this data of no significant relevance for our conclusions are thus we did not include them in tables. Nonetheless, we are willing to provide the data if deemed necessary.

  • -The discussion of the results should be more focused and directed towards the comparison of the current findings with previous studies.

We compared our results with those obtained in previous analyses of the literature and commented on potential differences in the Discussion section. Thank you.

  • -  What type of BS was performed? Are there differences between the two groups in terms of BS type?

We thank the reviewer for this observation. We acknowledge that different types of BS (i.e., gastric banding, bypass…) might influence the observed data on reported outcomes. All patients enrolled in the present research were subjected to sleeve gastrectomy at the Lircay clinic in Talca, Chile. Hence, no subgroup analysis based on the type of BS can be performed. We add this information to the methods section.

  • - Some parts of the text should be rewritten for clarity (for example lines 92-94 and lines 261-263). Moreover, the conclusion section should be focused on the main finding of the study (increased functional capacity in the training group) and do not extrapolate into other health parameters or morbidity risk.

We corrected English errors in mentioned lines and throughout the manuscript. Following your recommendations, we modified the conclusion section and the abstract. Thank you.

Minor point:

  • - BS has been shown to improve metabolic parameters and is considered as a therapeutic tool for diabetes remission. Have you studied the metabolic parameters of the patients included in your cohorts to find out if exercise had a positive influence?

This is an interesting aspect. We actually measured fasting blood glucose Unfortunately, we did not take into account other metabolites due to laboratory and budget limitations. We consider that glucose measures in isolation provide few insights into the patients’ metabolic profiles and might be out of the scope of the present manuscript.

Reviewer 2 Report

The work is well structured, but there are some things to review.

The novelty of the work is not clear,  what is the connection between functional capacity and bariatric surgery? I suggest stressing this point more, especially in the discussion.

moreover, even if the authors have put it as a limitation, the fact that there are more women can alter the results obtained, do the authors think that with an equal number of males the results would be different?

Author Response

The work is well structured, but there are some things to review.

We are thankful for receiving the reviewer’s comments on our manuscript.

The novelty of the work is not clear,  what is the connection between functional capacity and bariatric surgery? I suggest stressing this point more, especially in the discussion.

Bariatric surgery is a treatment that leads to weight loss and reduced fat accumulation which represent key factors that inference with the patient’s functional capacity. We tested if a physical exercise program can improve physical fitness hence further improving bariatric surgery results on functional capacity. We cited in discussion a meta-analysis which highlighted that the limited number of publications available to date represents a limitation that can hinder the applicability of exercise programs aimed to improved the outcomes of bariatric surgery and prevent cyclic weight gains. Our study expands on previous literature, particularly in patients subjected to sleeve gastrectomy. Thank you.

moreover, even if the authors have put it as a limitation, the fact that there are more women can alter the results obtained, do the authors think that with an equal number of males the results would be different?

As mentioned by the reviewer, a higher number of women might lower absolute values of covered distances observed in the functional capacity test. However, differences between groups and inside groups are not expected to differ with equal number of men since similar proportions of men and women were included in experimental and control group, as a sign of successful randomization of the participants.

Round 2

Reviewer 2 Report

The manuscript is ok .